# Magnetic and Viscoelastic Response of Magnetorheological Elastomers Based on a Combination of Iron Nano- and Microparticles

**DOI:** 10.3390/polym15183703

**Published:** 2023-09-08

**Authors:** Imperio Anel Perales-Martínez, Luis Manuel Palacios-Pineda, Alex Elías-Zúñiga, Daniel Olvera-Trejo, Karina Del Ángel-Sánchez, Isidro Cruz-Cruz, Claudia Angélica Ramírez-Herrera, Oscar Martínez-Romero

**Affiliations:** 1Institute of Advanced Materials for Sustainable Manufacturing, Tecnologico de Monterrey, Av. Eugenio Garza Sada Sur 2501, Monterrey 64849, Nuevo León, Mexico; daniel.olvera.trejo@tec.mx (D.O.-T.); kdelangel@tec.mx (K.D.Á.-S.); isidro.cruz@tec.mx (I.C.-C.); claudia.ramirezh@tec.mx (C.A.R.-H.); oscar.martinez@tec.mx (O.M.-R.); 2Tecnológico Nacional de México, Instituto Tecnológico de Pachuca, Carr. México-Pachuca Km 87.5, Pachuca 42080, Hidalgo, Mexico; palacios@itpachuca.edu.mx

**Keywords:** magnetorheological elastomer, nano- and microparticles, Mullin’s effect, crosslink density

## Abstract

In this paper, we discuss the creation of a hybrid magnetorheological elastomer that combines nano- and microparticles. The mixture contained 45 wt.% fillers, with combinations of either 0% nanoparticles and 100% microparticles or 25% nanoparticles and 75% microparticles. TGA and FTIR testing confirmed the materials’ thermal and chemical stability, while an SEM analysis determined the particles’ size and morphology. XRD results were used to determine the crystal size of both nano- and microparticles. The addition of reinforcing particles, particularly nanoparticles, enhanced the stiffness of the composite materials studied, but their overall strength was only minimally affected. The computed interaction parameter relative to the volume fraction was consistent with the previous literature. Furthermore, the study observed a magnetic response increment in composite materials reinforced with nanoparticles above 30 Hz. The isotropic material containing only microparticles had a lower storage modulus than the isotropic sample with nanoparticles without a magnetic field. However, when a magnetic field was applied, the material with only microparticles exhibited a higher storage modulus than the samples with nanoparticles.

## 1. Introduction

Magnetorheological elastomers (MREs) are materials that can have their mechanical and rheological properties modulated and controlled by external stimuli. Due to this, magnetorheological materials (MRMs) are heavily studied. The first investigations of suspensions on magnetorheological fluids (MRFs) date back to the 1980s. Silicone oil, greases, ionic liquids, and Boger fluids have been the most common viscoelastic media used to carry the magnetizable particles [1,2,3,4,5,6]. Suspensions may contain particles with various geometries, including fibers and spherical particles, with diameters ranging from microns to tens of microns [7], and mixtures of particles have been reported. For instance, Xu et al. [8] prepared multiwall carbon nanotube (MWCNT)-coated magnetic particles. The diameters of MWCNTs are in the range of 8–15 nm, and the magnetic particles have a diameter of 1.9 μm. The benefits of this modification were a decrease in the magnetic density of the particles and antisettlement. The sedimentation rate decreased as the viscosity increased, according to the magnetic particles’ volume fraction. Similar results were found by Choi et al. [9] when they studied the incorporation of MnFe_2_O_4_ nanoparticles and graphene oxide (GO) nanosheets into carbonyl iron particles (CIPs) as fillers of MRFs. They observed slower sedimentation due to the dispersion stability attributed to GO sheets’ low density, planar structure, and resistance to MnFe_2_O_4_ attached over the surface of magnetic particles. Also, the performance of nanoparticles (NPs) as a unique filler has been investigated. Rwei et al. [10] studied the influence that Fe_3_O_4_ NPs, with a mean size of 10 nm, have on magnetorheological suspensions. They found that adding NPs into the fluid solution increased the MR effect based on the intensity of the applied magnetic field. They also found that adding less than 1 wt.% Fe_3_O_4_ produced an increment of up to two orders of magnitude of the yield stress in the MR material. Mixtures of different sizes of the same type of particles have been investigated by Park et al. in [11]. They observed higher yield behaviors in the MR fluids made from a combination of micro and nanomagnetic particles. The MR fluids response was attributed to good alignment and reinforcement of nano and microparticles during the magnetic field application. Up to now, the study of MR suspension fluids reinforced with micro and nanomagnetic particles has received significant attention; however, this is not the case for magnetorheological materials in their solid state. Figure 1 summarizes the number of published works on MRMs from 1979 until the present. Notice from Figure 1 that the common particles used to manufacture MREs are carbon iron particles (CIPs) with diameter sizes in the range of 3.5–5 μm [12,13,14,15]. It can also be seen from Figure 1 that only a few articles used nanoparticles to investigate MRE behavior, such as the article published by Bica [16]. In his study, Bica obtained iron nanoparticles through a microwave process; then, he attached the developed NPs to a polymeric matrix surface using a dielectric-based material of a planar capacitor. A few publications have recently reported mixing formulations of micro and nanoparticles added into elastomer materials to investigate their effect on the mechanical and rheological properties of the composite magnetorheological elastomer materials. 

For instance, Lee et al. [17] added γ-Fe_2_O_3_ nanoparticles and CIPs to a natural rubber composite elastomer. They observed higher storage modulus values for those hybrid magnetorheological elastomers for increasing magnetic fields. They attached rod-shaped γ-Fe_2_O_3_ nanoparticles over the CIPs’ surface and added nanoparticles to the space among the micro iron particles in an attempt to stabilize the rubber material chain. Bica et al. [18] reported the performance of plane capacitors manufactured from hybrid magnetorheological elastomers based on silicone rubber, carbonyl iron, and graphene NPs. They found an increment in capacitance with the increasing magnetic field intensity influenced by the nanoparticles’ concentration. Khayam et al. [19] used iron micro and nanoparticles to manufacture MREs. They observed that nanoparticles showed a remarkable role in increasing the MR effect in hybrid MREs compared with conventional MREs. They found that the material stiffness is modified depending on the wt.% and the sizes of the added micro and nanoparticles.

Previous works have provided no evidence of the performance of hybrid magnetorheological elastomers simultaneously subjected to uniaxial loads magnetic fields in situ. Therefore, the magnetorheological effects of mixing micro and nanoparticles in an elastomer matrix material need to be investigated.

This article focuses on developing a magnetorheological elastomer reinforced with mixes of nano and microparticles and studying the influence that these have on the material mechanical properties when subjected to a magnetic field.

## 2. Materials and Methods

### 2.1. Materials

The MREs samples were made from a matrix of polydimethylsiloxan (PDMS) by mixing two components—part A and part B, from Ecoflex with a viscosity of 14,000 cps according to ASTM D-2393 [20] and shore hardness of 00-10 according to ASTM D-2240 [21]. Dimethyl-silicone oil (SO), whose viscosity was 0.25 Pa·s, was used as an additive to coat the surface of the magnetic particles. We purchased PDMS resin and SO from Mörph Industries (México City, México). Carbonyl iron particles (CIPs) with particle sizes from 5 to 9 μm, and iron nanoparticles ranging from 60 to 80 nm were used as fillers. Both fillers were purchased from Sigma-Aldrich (México City, México). Toluene purchased from Jalmek Científica (Monterrey, México) was used for swelling tests.

### 2.2. Procedure for Manufacturing the MREs

The MREs were made from a PDMS matrix, mixing in equal proportions (1:1) of part-A and part-B to initiate the curing process. As a result, the mixture changed from a viscous liquid to a solid when the curing process ended after 4 h. The manufacturing process used to produce the MREs is described in detail in [22]. For convenience, we would like to briefly describe some steps we have followed to produce the magnetorheological material samples. First, the CIPs powders were mixed with the silicone oil (SO) for a few minutes. Then, the NPs powders were added and mixed again until all the particles were covered with SO, and a homogeneous mixture was obtained. We named this mixture Mix1. Next, parts A and B of the PDMS resin were poured into separate beakers following a weight relation of 1:1. Then, part-B was mixed with mixture Mix1 for a few minutes until we obtained a homogenous mixture named Mix2. Finally, part-A was poured into the previous blend (Mix2) and remixed to obtain a homogenous mix named Mix3. Immediately, Mix3 was poured into the molds to obtain the different specimens used for rheological and uniaxial extension tests. Then, the molds were put under a vacuum media to eliminate the air encapsulated inside the mixture during the stirring process. The samples were made with 45 wt.% of magnetic particles, using two different percentages of nano and micro sizes: 0% of nanoparticles with 100% of microparticles (identified as 0 N 100 M), and 25% and 75% of nano and microparticles, respectively (identified as sample 25 N 75 M). The remaining mixture, 55 wt.% corresponds to the PDMS resin and SO (24 wt.%). Since the content of magnetic particles for all the samples was the same (45 wt.%), the amounts of PDMS and SO were also the same for each produced sample. During the first 30 min of the curing process, a magnetic field with 52 mT was applied to induce the alignment of the magnetic particles that allowed anisotropic MREs samples to be obtained. For the isotropic MREs samples, the experimental procedure and content of fillers used were the same as those used to manufacture the anisotropic samples. However, we did not apply a magnetic field during the sample curing process. Lastly, all samples were unmolded after a curing time of 8 hours. The PDMS resin used in this study possesses low viscosity that allows the addition of higher wt.% of filler. However, we observed that when NPs were used, the high surface nanoparticles area limited the combination with microparticles to a maximum ratio of 25:75 (25 N 75 M).

### 2.3. Characterization Methods

The particle size for the nano and micro iron particles was analyzed via scanning electron microscopy (SEM). CIPs microparticles were observed using a SEM, Evo MA 25, Carl Zeiss (Oberkochen, Germany). To measure the iron nanoparticle size, the nanoparticle powder was dispersed in isopropyl alcohol, and then a maximum of two drops were deposited on a copper grid, allowing sufficient time for alcohol volatilization. Finally, the iron nanoparticle size was analyzed using a SEM Quanta 250-FEG FEI (Eindhoven, The Netherlands).

The crystalline structure of the magnetorheological solid samples was identified through X-ray diffraction (XRD) analysis using a PanAnalytical X’Pert Pro PW1800 diffractometer configured to deliver a scanning rate of 2°·min^−1^ by means of Cu Kα radiation (Almelo, The Netherlands). The system was controlled at 45 mA and 40 kV, and the measurements were performed from 10° to 150° in 2*Ɵ*. The crystal size of magnetic particles was determined using the Debye Scherrer Equation: (1)$$D=\frac{K\lambda }{\beta \cos\theta }$$

This mathematical expression is widely used to compute the crystal size of several particles [19,23,24]. Here, K is the Scherrer constant (0.94), λ is the wavelength of the X-rays used ($\lambda_{\mathrm{Cu}}$ = 1.54 Å), θ is the peak position (radians), and D is the crystallite size. A Gaussian fit of the most intense peak of the diffractograms was performed to calculate the full-width–half-maximum (FWHM, β). Fourier-transform infrared spectroscopy (FTIR) (PerkinElmer Frontier equipped with a universal attenuated total reflectance (UATR) accessory, Waltham, MA, USA) was used to identify functional groups in the polymer matrix. Each measurement was performed in the middle interval range of 4000 to 400 cm^−1^ at 8 cm^−1^ of resolution with 16 scans on average.

The thermal stability of polymeric samples was evaluated by heating each sample from room temperature to 800 °C at a heating rate of 10 °C·min^−1^ and nitrogen gas flow of 20 mL·min^−1^ using a PerkinElmer thermogravimetric analyzer equipped with the software Pyris 1. The density of crosslink chains was determined through a swelling test that exposes the polymer material to an unaggressive solvent that allows the swelling through the elongated polymer chains without degradation. Initially, 10 × 4 × 2 mm specimens were weighed and plunged into toluene at room temperature and darkness. The solvent was replaced every 24 hrs to reduce interference from toluene-soluble residues in the polymer materials [25] until a lapsing time of 72 h. Then, the swollen samples were filtered and weighed again. Finally, the specimens were dried at 80 °C for 24 h using a drying oven; then, these samples were weighed at room temperature to obtain a reliable weight. For each specimen, the swelling test was conducted three times. Based on the experimental data, the volume fraction ($V_{r}$) was computed using the following Equation:(2)$$V_{r}=\frac{V_{p}}{V_{p}+V_{s}}=\frac{\frac{m_{dry}}{\rho_{r}}}{\left(\frac{m_{dry}}{\rho_{r}}+\frac{m_{wet}-m_{dry}}{\rho_{s}}\right)}$$
where $m_{dry\;}$ and $m_{wet}$ are masses of the dried and swollen specimens, respectively, $\rho_{r}\;$ indicates the density of PDMS rubber (1.1 g·cm^−3^), $\rho_{s}$ stands for the solvent density (for toluene is 0.865 g·cm^−3^). As reported in [26], the shear modulus μ is obtained from tensile tests using Treolar’s relationship [27]
(3)$$\mu =\frac{\rho_{r}}{M_{c}}RT=\left[X\right]RT$$
Here, $\left[X\right]$ is the material’s crosslinked density, $M_{c}$ represents the mean chain molecular weight between successive points of cross-linkage, R corresponds to the molar gas constant, and T is the temperature in °K. According to the Flory–Rehner equation
(4)X=−ln1−Vr+Vr+χVr2V0Vr13−Vr2
the solvent and the composite material interaction parameter χ is determined using the polymer volume fraction $V_{r}$, extracted from the swelling test and the crosslink density $\left[X\right]$ computed from Equation (3). Thus, uniaxial experimental tests at room temperature were performed on the MREs dumbbell-shaped (ISO37-2011 [28]) samples using the Instron 3365 universal testing machine. The MRE experimental stiffness value was determined by subjecting the material samples to a maximum elongation value of 20 mm at a crosshead rate of 200 mm·min^−1^.

MREs’ viscoelastic behavior was measured using an advanced commercial rheometer (Model: MCR301, Anton Paar, Monterrey, México) with a parallel-plate rotor that oscillates at a specific frequency. A magnetic device induces the applied magnetic field, and a metal cover enhances and guides the magnetic field perpendicular to the surface of the MRE sample. Experimental dynamic tests under controlled frequency and strain were performed on cylindrical samples of 10 mm and 1 mm in diameter and thickness, respectively. All measurements were performed at 20 °C.

## 3. Results

### 3.1. Chemical Analysis by FTIR Spectroscopy

Figure 2 shows the recorded FTIR spectra. Both bare and hybrid nano-microparticle samples were analyzed. The results show the characteristic peaks of the polydimethylsiloxane matrix in the fingerprint region. Stretching vibrations of Si-O-Si of crosslinked PDMS appear at 476, 1010, and 1079 cm^−1^ [29,30]. A sharp peak at 789 cm^−1^ corresponds to the asymmetric stretching and bending vibration modes of Si–H. Also, a tiny band at 864 cm^−1^ is attributed to Si-OH bonds [29,31,32]. At the value of 1257 cm^−1^, the Si-C in Si-CH_3_ stretching vibration mode is found [33]. Finally, the short peaks at 1412, 1445, and 2905–2962 cm^−1^ correspond to Si–CH=CH_2_ vibration, C-H bending in -Si-CH_2_, and CH_2_ stretching in Si-CH_2_- [31,34]. No chemical interaction was observed, since the manufacturing process was performed at room temperature. In this research, the FTIR results do not show other additional bands to the PDMS matrix. These results agree with those reported in Ref. [30], in which the same analysis by FTIR was performed along with XRD and X-photoemission spectroscopy (XPS) to detect chemical material modifications. However, our measurements confirmed that there were no material modifications due to the interaction of the elastomer matrix with the micro and nanoparticles.

### 3.2. Crystallographic Analysis by XRD

The XRD measurements shown in Figure 3a demonstrate the crystalline nature of the nano and the microparticles. Sharp peaks located at 44.6, 65, 82.3, and 98.9 in 2θ are linked to the α-ferrite phase with the cubic structure according to JCPDS 06-0696, being the crystallographic plane *hkl* (110) the most crystalline. Figure 3b confirms the appearance of the crystalline phase represented by the intense peaks (44.6° to 116° in 2θ) and the amorphous phase due to polymeric material located from 10–25° in 2θ. Since all the magnetorheological materials were manufactured to 45 wt.% of magnetic particles, all samples exhibit the same intensity for each crystalline plane. However, the incorporation of nano and microparticles does not affect the chemical nature of the polymeric matrix since the nano and microparticles’ content does not modify the intensity of the main bands of the different functional groups observed during FTIR analysis. The inset of Figure 3a shows the maximum peak from which the FWHM β is determined, giving, as a result, crystal sizes for micro and nanoparticles of 64.6 and 43 nm, respectively.

### 3.3. Morphological Analysis by SEM and Optical Microscopy

SEM measurements were conducted to evaluate iron nano and microparticle morphology and size. Using several SEM images, it was possible to measure the particle size statistical distribution using Digimizer 4.6.1. Figure 4 depicts spherical particles for both particle sizes with some nanoparticles’ irregular morphologies. The size distribution observed in Figure 4 indicates an average size of 70 nm and 1.25 μm for nano and micro CIPs particles, respectively. The crystal size determined using the Scherrer equation agrees with the particle size obtained by SEM analysis, which means that the particles are formations of smaller crystals. An approximation was computed to estimate the number of crystals that form a particle, considering the particle size and crystal size. Results indicate that 3793.4 crystals form the microparticles, and only 2.27 crystals form the nanoparticles. A study by Manglam et al. [35] found similar results for the particle and crystal size of barium hexaferrite particles. They realized that the particles were bigger than the crystallites measured by XRD patterns, which they attributed to the usual characteristics of oxide samples. Figure 5 shows the distribution of the magnetic particles into the PDMS matrix for the sample Ani 25 N 75 M. Notice the magnetic particle alignment through the straight lines that the magnetic particles form in the anisotropic samples due to the magnetic field applied during the curing process. However, some clusters of particles are also observed from the image recorded via optical microscopy. These clusters have an approximate size of less than 10 μm. It is worth mentioning that at this scale, it is impossible to distinguish between the micro and nanoparticles because of the constrained resolution of the optical microscopy. To see the particle distribution in the polymeric matrix, we would like to refer the interested reader to some previously published articles [22,26,30,36] in which additional details about their morphology, average size, and distribution are provided.

### 3.4. Thermal Stability by TGA

Figure 6 shows the magnetic particles’ influence on the material thermal stability through the TGA measurement analysis. When compared with the bare material, an increment of thermal stability was observed for each sample reinforced with both nano and microparticles. From the TGA and DTGA curves shown in Figure 6a,b, the thermal stability parameters, such as the 10% weight loss temperature (T_10%_), the maximum peak decomposition temperature (T_p_), and the peak decomposition rate (R_p_) could be obtained. These values are listed in Table 1.

The thermal stability of the cured PDMS matrix was improved with the addition of nano and micromagnetic particles. T_10%_ of unfilled cured PDMS was identified at 365 °C, and a significant thermal stability increment for the materials Iso 0 N 100 M (T_10%_ = 401 °C) and Ani 0 N 100 M (T_10%_ = 399 °C) was observed. From Figure 6b, we can identify three major maxima weight loss temperatures for unfilled PDMS bare material. The first maximum peak decomposition temperature (T_p1_) appears at 414 °C, where ~27.5% of the polymer was eliminated. The second peak decomposition temperature, T_p2_, was measured at 478 °C, where ~47% of cured PDMS was burned. Finally, approximately 52% of the polymeric matrix was decomposed at 510 °C (T_p3_). The corresponding maxima weight loss temperatures of reinforced materials remarkably increased. The first maximum degradation step (T_p1_) was shifted from 424 to 439 °C, while the second maxima weight loss temperature was incremented from 478 °C to 512–602 °C. Notice that in the first maxima weight loss temperature (T_p1_), the higher increment was observed for the Ani 25 N 75 M material, but the higher increase for the T_p2_ was for the Ani 0 N 100 M sample. See Table 1. Reinforced materials did not exhibit a third maximum degradation peak (T_p3_). A thermal stability increase of elastomeric materials reinforced with magnetic particles was also found by Wu et al. in [37]. The highest peak decomposition rate (R_p_) occurs at T_p1_ for all material samples. The slight weight increment observed in the magnetorheological elastomer Ani 25 N 75 M can be attributed to the possible oxidation of the iron particles when nitrogen is used to perform the experimental measurements. The same finding was reported in [38] when nitrogen gas was used. Because of this, an increase in residues is observed. The residue percentage for the PDMS bare sample is 36.8%, whereas for the enhanced materials, it ranges from 61 to 71%. The residues contain higher amounts of mass than expected since only 45 wt.% of iron particles were added to the material. This could be attributed to the presence of crosslinked structures that are resistant to decomposition even at temperatures close to 800 °C. Moreover, the residues could contain iron due to its high melting temperature, which is over 1530 °C.

### 3.5. Uniaxial Extension Results

Uniaxial extension tests were performed on PDMS bare elastomer specimens and in the composite ones. Figure 7 shows that the reinforced samples with filling material exhibit an increment in the material stiffness but a reduction in the ultimate stretch values. Also, a marginal increment in the tensile strength is observed.

Figure 8 details the composite materials’ performance in comparison with the bare sample. The stiffness of the composite material increases with the presence of filling particles, which is more evident when the filler material is aligned inside the polymeric matrix, i.e., the anisotropic samples. In particular, the anisotropic material with 25% nanoparticles and 75% microparticles (Ani 25 N 75 M) increases the stiffness by 4.5 times in comparison with the bare sample. In the case of isotropic materials, their stiffness is more significant in materials with nano and micro particles than in those with only microparticles added. The latter was also observed for the anisotropic samples. Also, it can be seen that the presence of nanoparticles decreases the ultimate stretch and the tensile strength values of the material samples.

### 3.6. Rheological Test Results

This section shows the material magnetorheological performance of the produced magnetorheological material samples. Figure 9 depicts the evolution of the storage modulus when subjected to oscillatory frequency values in the interval of 0.01 to 100 Hz. This test was performed with a magnetic field intensity of 1 T, and for comparison, the same test was executed without the application of the magnetic field. In all cases, there was a significant increment in the storage modulus when the magnetic field was applied. Figure 9a,b show that the anisotropic material possesses a better magnetorheological effect at low frequencies, while the isotropic material exhibits a better magnetorheological effect at high frequencies. There is a significant change in the storage modulus for specimens without nanoparticles, as illustrated in Figure 9c,d. It is essential to mention that all curves shown in Figure 9a–d were plotted from experimental data collected with a constant shear strain of γ = 1%, with the parallel plates subjected to an average normal force of 1.5 N.

One can see from Figure 9c that the anisotropic samples containing nanoparticles are consistently stiffer than the ones with only microparticles across all frequency ranges without a magnetic field. However, when a magnetic field of 1 T is applied, the composite material with magnetic nanoparticles displays a more significant magnetorheological effect at frequencies above 30 Hz than with only microparticles.

Figure 10 illustrates the variation in the storage modulus measured when the magnetic field is increased from 0 to 1 T. These measurements have been performed at a constant frequency value of 10 Hz, a shear strain value, γ, of 1%, and a normal force value of 1.5 N. One can see from Figure 10 that the material samples containing only microparticles as a filler (Ani 0 N 100 M and Iso 0 N 100 M) have higher storage modulus values when a magnetic field of 1 T is applied. Samples using nanoparticles and microparticles as fillers (Ani 25 N 75 M, Iso 25 N 75 M) exhibit a higher storage modulus when no magnetic field is applied. Comparing the sample’s performance Ani 0 N 100 M (red line) with the one identified as Ani 25 N 75 M (yellow line), we can observe a larger storage modulus value in the material samples with nanoparticles up to a magnetic flux intensity value of around 100 mT. When this magnetic flux density value is exceeded, the samples with only microparticles exhibit a greater storage modulus value. For the case of the samples Iso 0 N 100 M (Blue line) and Iso 25 N 75 M (Green line), the material samples with nanoparticles exhibit a superior storage modulus value from 0 mT up to 400 mT, and then, the material samples with microparticles only have a better storage modulus value.

Figure 11 illustrates the Cole–Cole plots of the composite samples with the loss and the storage moduli plotted on the horizontal and vertical axes, respectively. Notice from Figure 11a that there is a difference in the behavior of the anisotropic and isotropic samples with only microparticles added when a magnetic flux density of 1 T is applied. An increase in loss and storage modulus is observed, regardless of the frequency used in the test. Figure 11b displays similar behavior for the anisotropic and isotropic samples with nanoparticles added. Figure 11c compares anisotropic samples, while Figure 11d depicts a comparison for the isotropic ones. It is worth mentioning that the sample with only microparticles has lower storage modulus values compared to the sample with nanoparticles when a magnetic field is not applied. However, when the magnetic field is on, the material samples with microparticles exhibit a higher storage modulus compared to those with nanoparticles.

### 3.7. Swelling Test Results

Data from swelling and tensile tests could be used to evaluate the behavior of PDMS elastomers reinforced with magnetic particles using Equations (3) and (4). In Equations (3) and (4), the crosslinking density is determined from tensile experimental data, and the volume fraction of PDMS is measured through a swelling test. The interaction parameter value is then computed using the Flory–Rehner equation. In Figure 12, the blue stars represent the interaction parameter values obtained from Equation (4). As expected, our computed results show good agreement with those obtained by Chahal [39] (blue circles) and by Schuld [40] (red diamonds). It is also evident that the interaction parameter depends on the polymer volume fraction, making swelling and tensile tests helpful for determining the crosslinking properties of the developed MRE composites. The crosslinked density increases for all composite materials, but the highest increment is observed in the samples reinforced with nanoparticles, as shown in Table 2. The value for the volume fraction ($V_{r}$) for the bare sample is 0.14, and for the composites elastomers, its value is in the range from 0.325 to 0.346.

## 4. Conclusions

The investigation focused on the amalgamation of nano and microparticles within a PDMS matrix to scrutinize the impact of magnetic particles on material behavior attributes. The outcomes elucidate the formation of a composite material characterized by both amorphous and crystalline phases, corresponding to the polymeric matrix and metallic particles. As expected, the chemical structure of the polymer remains unaffected by the incorporation of magnetic particles of varying sizes. Intriguingly, the presence of magnetic particles bolsters the thermal stability of the polymeric matrix. Through SEM analysis, we identified the nanoparticles and microparticles’ size and morphology provided by the supplier. The use of the Scherrer equation aided in the computation of crystal sizes. Tensile testing provided valuable information regarding the stiffness enhancement of the composite material. This improvement in the material stiffness is particularly pronounced when adding iron nanoparticles. Moreover, the effects of adding nano and microparticles in relation to the material behavior can be qualitatively and quantitatively observed in the performed tensile tests. In summary, this investigation unveils a heightened magnetic response beyond 30 Hz in composite materials reinforced with nanoparticles. Interestingly, the isotropic material exclusively containing microparticles exhibits a lower storage modulus in the absence of a magnetic field. Yet, the scenario shifts under the influence of a magnetic field, as the material with only microparticles demonstrates a superior storage modulus when compared to samples containing nanoparticles. These findings provide valuable information that provides us with a deeper comprehension of the intricate interplay between magnetic particles and composite material properties.

## Figures and Tables

**Figure 1 polymers-15-03703-f001:**
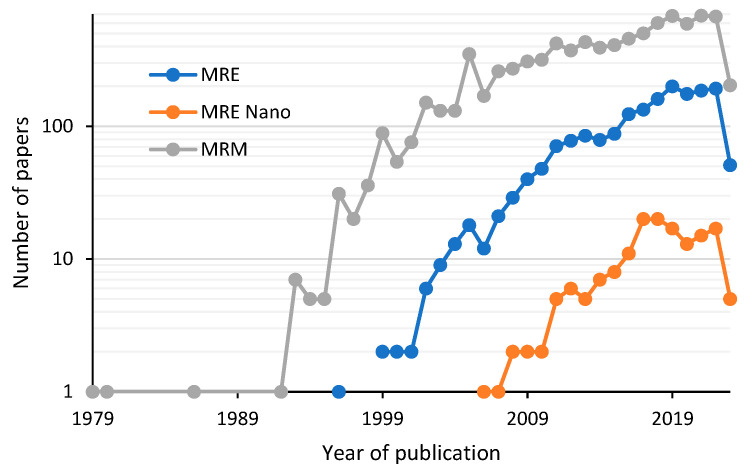
Publications reported by Web of Science. The gray line represents all the papers that deal with magnetorheological materials (MRMs) from 1979 until this year. The blue line indicates the studies performed on magnetorheological elastomers regardless of the kind of reinforced material. Finally, the orange line denotes only those papers that investigate magnetorheological elastomers manufactured with nanoparticles (MRE Nano). The vertical axis is on a log scale.

**Figure 2 polymers-15-03703-f002:**
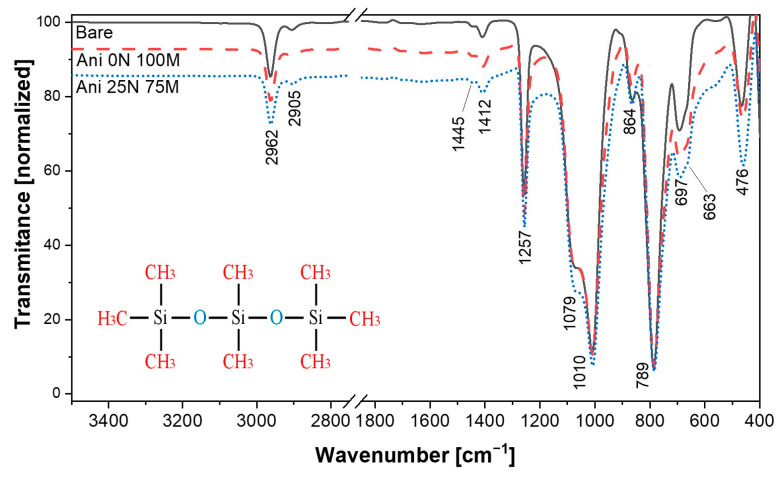
FTIR spectra of magnetorheological samples manufactured to different combinations of nano and microparticles.

**Figure 3 polymers-15-03703-f003:**
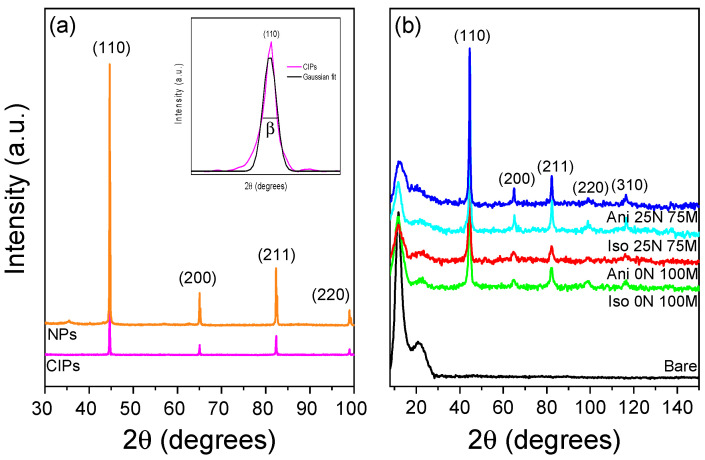
Diffractograms of (**a**) nano and microparticle powder and (**b**) different ratios of iron nano/microparticles as fillers of PDMS-based magnetorheological materials. The inlet figure represents a zoom-in for the (110) crystallographic plane of CIPs, and its Gaussian fit was applied to find the full width at half maximum (FWHM), denoted as β, to determine crystalline size using the Scherrer equation.

**Figure 4 polymers-15-03703-f004:**
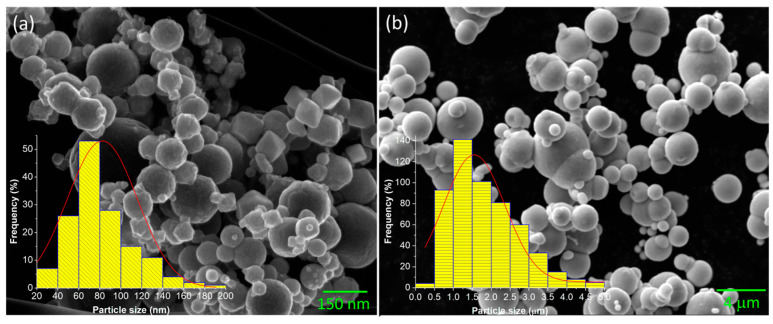
Size distribution and micrography of magnetic particles: (**a**) nanosize and (**b**) microsize recorded by SEM analysis.

**Figure 5 polymers-15-03703-f005:**
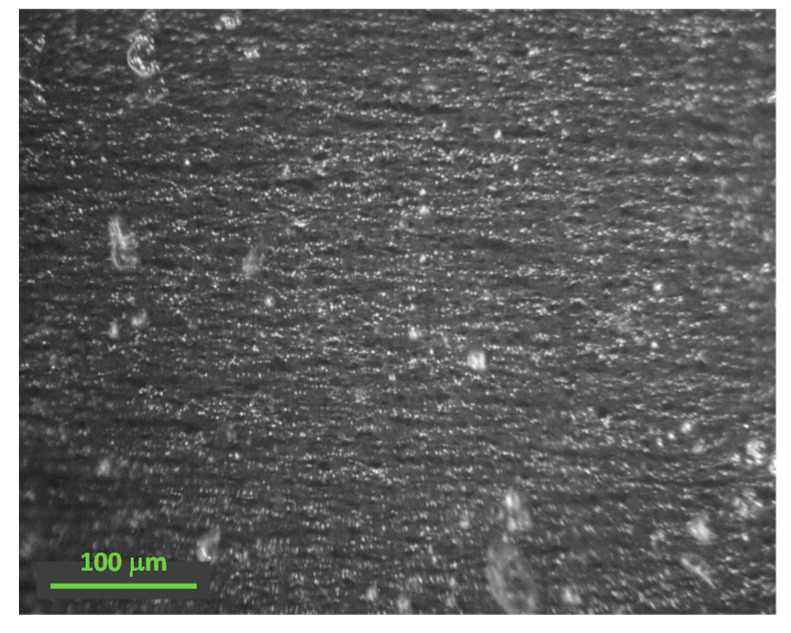
Optical microscopy image that shows the magnetic particle alignment into the PDMS matrix of the sample Ani 25 N 75 M.

**Figure 6 polymers-15-03703-f006:**
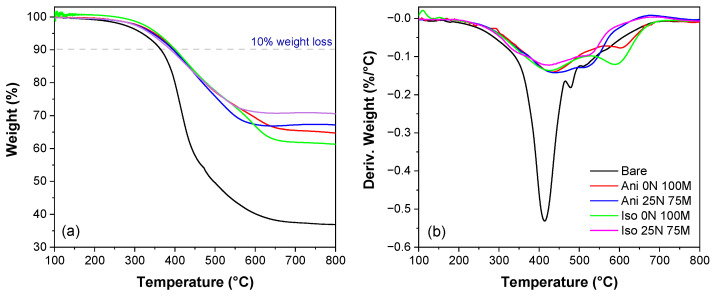
(**a**) TGA analysis and (**b**) DTGA curves used to investigate the influence of the combination of nano and microparticles in the material thermal stability.

**Figure 7 polymers-15-03703-f007:**
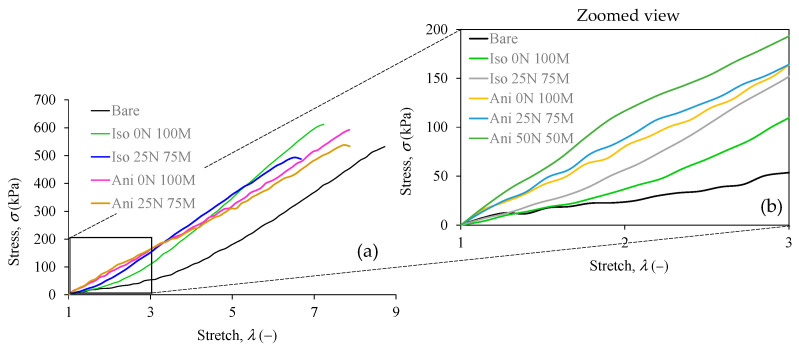
Stress versus stretch response curves were obtained from the bare polymeric matrix sample and the composite material samples reinforced with nano and microparticles. Notice that the material stiffness increases with the presence of filling material. These curves were used to find the sample’s shear modulus and the crosslink chain density using Equations (3) and (4) along with the swelling test data. (**a**) General view of the tensile test curves, (**b**) a zoomed view in the zone of low stretch.

**Figure 8 polymers-15-03703-f008:**
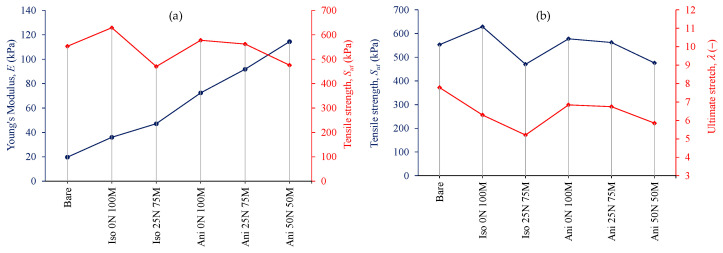
(**a**) Young’s modulus and tensile strength and (**b**) Tensile strength and ultimate stretch for the bare and composite material obtained from the tensile test. All samples have 45% wt. of magnetic particles.

**Figure 9 polymers-15-03703-f009:**
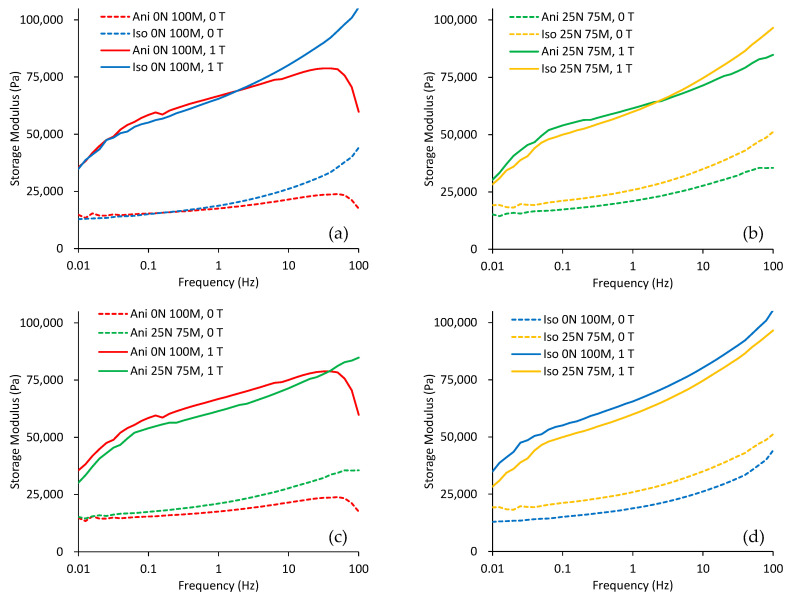
Parallel plate rheological test in the frequency domain. The strain is kept constant at γ=1%, with a constant normal force of 1.5 N and a constant field intensity of 0 T and 1 T. (**a**) Samples with only microparticles, (**b**) samples with nano- and microparticles, (**c**) anisotropic samples, and (**d**) isotropic samples.

**Figure 10 polymers-15-03703-f010:**
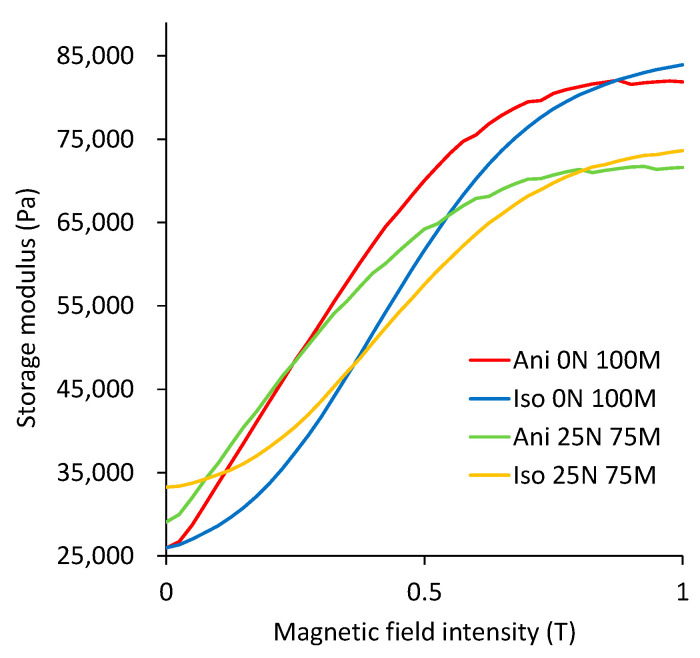
Storage modulus versus magnetic field. Notice that at higher magnetic fields, the isotropic specimens made with iron microparticles attained the highest storage modulus value.

**Figure 11 polymers-15-03703-f011:**
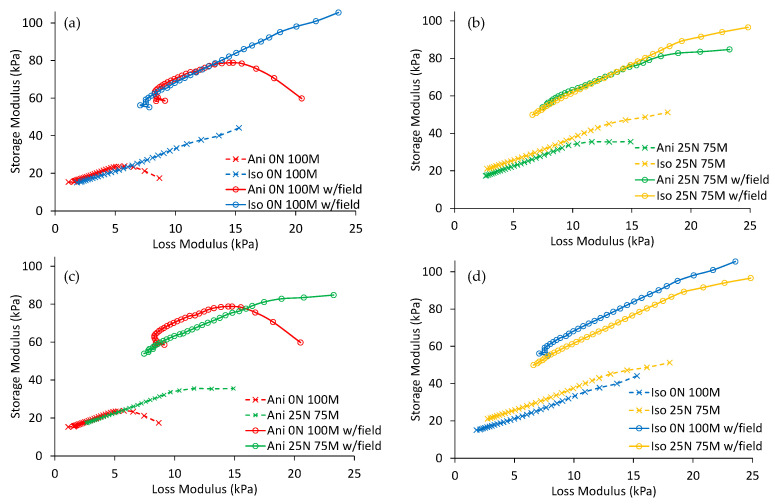
Cole–Cole plots showing the influence of the magnetic flux density on the composite materials. (**a**) Samples with only microparticles, (**b**) samples with nano- and microparticles, (**c**) anisotropic samples, and (**d**) isotropic samples.

**Figure 12 polymers-15-03703-f012:**
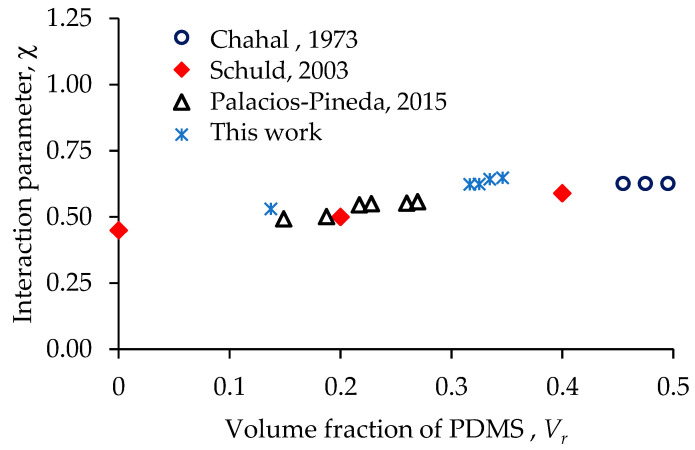
Interaction parameter values obtained from swelling tests as a function of the volume fraction. Comparison with the data from Chahal [39], Schuld [40], and Palacios-Pineda [24].

**Table 1 polymers-15-03703-t001:** Thermal stability parameters recorded from TGA and DTGA curves.

Sample	T_10%_ (°C)	T_p1_ (°C)	T_p2_ (°C)	T_p3_ (°C)	R_p1_ (%/°C)	R_p2_ (%/°C)	R_p3_ (%/°C)	Residue (%)
Bare	365	414	478	510	0.531	0.186	0.125	36.8
Iso 0 N 100 M	401	427	589	---	0.136	0.12	---	61.3
Ani 0 N 100 M	399	432	602	---	0.14	0.076	---	64.7
Iso 25 N 75 M	389	424	516	---	0.122	0.098	---	70.6
Ani 25 N 75 M	396	439	512	---	0.141	0.128	---	67.2

**Table 2 polymers-15-03703-t002:** Volume fraction, crosslinking density, and interaction parameters computed from the experimental data of swelling tests.

Material	[X] [mol/m^3^]	*V_r_* [-]	χ [-]
Bare	8.1	0.137	0.531
Ani 0 N 100 M	29.7	0.317	0.623
Ani 25 N 75 M	37.7	0.325	0.624
Iso 0 N 100 M	14.8	0.335	0.643
Iso 25 N 75 M	19.3	0.346	0.648

## Data Availability

The data presented in this study are available on request from the corresponding author.

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
