# Peer review of "Magnetic and Viscoelastic Response of Magnetorheological Elastomers Based on a Combination of Iron Nano- and Microparticles"

_polymers, 2023, doi:10.3390/polym15183703_

Round 1
Reviewer 1 Report
The manuscript entitled " Magnetic and viscoelastic response of magnetorheological elastomers based on a combination of iron nano and microparticles” written by I. Anel Perales-Martínez et al., was submitted to Polymers.
The manuscript report on magnetorheological properties of hybrid elastomers prepared by authors. Elastomers based on polydimethylsiloxane (PDMS) were admixed with the carbonyl iron particles (CIPs) and the iron nanoparticles.
At first, the motivation was not explained properly. What can be expected when authors combine micro and nanosized particles? In what aspect would it be valuable, what does it mean “a better understanding” of such a system?
In materials and procedures, the explanation is not in ideal form concerning the preparation of isotropic sample and other one. The difference is in application of the magnetic field during the curing process?
If both fillers }magnetic nano and microparticles) were acquired commercially by Sigma-Aldrich, I do not see any reason to show the SEM photos (Figure 4 is not the result of this contribution).
In Figure 6 and most of other figures, I do not a difference between isotropic and anisotropic samples. Characterization methods are properly described. Nevertheless, the FTIR spectra do not show clear change in properties when comparing the bare elastomer and hybrid magnetorheological elastomer (MRE).
Authors found that the stiffness of the composite material increases with the presence of filling particles. This result is more evident when the filler material is aligned inside the anisotropic samples. In any case, it is not clear how it is relevant to the nanoparticle size or composition, etc.
Authors concluded that magnetic particles and their sizes do not affect the chemical structure of this type of magnetorheological elastomer. It is slightly strange as no systematic study has been presented. Additionally, authors explicitly admit that they observed “the strength and the ultimate stretch follow the same pattern…”. The conclusion that the presence of magnetic particles increases the thermal stability of the elastomers is more-less general and universal. More data and additional information should be added into the original manuscript.
Generally, the results look incremental and the characterization slightly routine. Other system preparation and /or additional studies are necessary to draw valuable conclusions. I recommend to reject the paper, I encourage the authors to complete their experimental work.
Reviewer 2 Report
My comment:
1) Giving information about background is good but this is a research paper and as far as I know, it is not conventional to present a diagram like Fig.1 in the research paper, it is more suitable for a review paper so remove it
2) In the material process section, please clarify the process of the material with more detail and the benefit and drawbacks of the material clearly
3) For XRD and measurement technique please cite related paper for Eq.1
4) Don’t use the solid lines for all diagrams in a Figure such as Fig.2, in addition, increase the width of lines and fonts for increasing the quality of images
5) Fig.4 caption is placed on the next page, and also about this sentence please give reference “Scherrer equation is according to particle size obtained by SEM analysis”.
6) Why we can see a distortion in Fig.5 (b) for the bare surface more details should be presented for Fig. 5 (a) and (b)
7) For Fig.6, it looks like Eq.4 is used for the tensile strength based on the Flory-Rehner equation, if yes so please mentioned it in the description of the image and highlighted it for readers
8) In Fig.8, some abnormal differences between Ani and ISO can be seen at 100 Hz in Fig.8 (a) and also Ani 0N and 25 N in Fig.8 (c), if it has a technical reason.
in brief, your paper is very good and I enjoy it because has a very clear process and description. I think only you should mention some abnormal changes and some comparing with other published papers in this field
Reviewer 3 Report
Dear Editor,
The authors presented their studies on the mechanical properties of magnetoelastomers based on the iron particles dispersed in the PDMS polymer matrix. The data for the influence of nano and micron size iron particles are presented. The experimental procedure and the characterization methods are described are described well enough. The conclusion has been improved. For example "SEM analysis enabled us to determine the size and morphology of the nanoparticles and microparticles, and the Scherrer equation was used to calculate crystal size." - it would be more appropriate to be in the results section. The Discussion section is messing. It is not clear the particle distribution in the polymer matrix. Are the particles dispersed as single one or form clusters. Please, show the SEM images for magnetoelastomer.
Best regards
Round 2
Reviewer 1 Report
I can recommend the manuscript for publication.
Author Response
We thank the reviewer for his/her valuable comments that helped us improve a previous version of our submitted manuscript.
Sincerely yours
Alex Elías-Zuniga, Ph.D.
Reviewer 2 Report
the paper is revised carefully and it can be published
Author Response

(The authors gave the same response as above.)

Reviewer 3 Report
Dear Editor,
The authors did appropriate changes in the manuscript, and I suggest to be publish.
Best regards
Author Response

(The authors gave the same response as above.)
